# Source Depth Discrimination Using Intensity Striations in the Frequency–Depth Plane in Shallow Water with a Thermocline

Xiaobin Li [1,2] and Chao Sun [1,2,*]

1   School of Marine Science and Technology, Northwestern Polytechnical University, Xi'an 710072, China; xblee@mail.nwpu.edu.cn
2   Shaanxi Key Laboratory of Underwater Information Technology, Xi'an 710072, China
*   Correspondence: csun@nwpu.edu.cn

**Abstract:** A source depth discrimination method based on intensity striations in the frequency–depth plane with a vertical linear array in a shallow water environment is proposed and studied theoretically and experimentally. To quantify the orientation of the interference patterns, a generalized waveguide variant (GWV) $\eta$ is introduced. Due to the different dominance of the mode groups, the GWV distribution in the surface source is sharply peaked, indicating the presence of striations in the interferogram and the slope associated with the source–array range, while the distribution of the submerged source is more diffuse, and its interferogram is chaotic. The existence or lack of a distinct peak is used to separate the surface and submerged source classes. The method does not demand prior knowledge of the sound speed profile or the relative movement between the source and the array. In addition, it is the presence of the striations, not the value of $\eta$, that is exploited to separate the surface and submerged source classes, which means the source–array range can be unknown. The proposed method is validated using experimental data on the towing ship in SWellEx–96 and numerical modeling. The method's performance under noise situations and for different source–array ranges is also investigated.

**Keywords:** source depth discrimination; modal interference in frequency–depth plane; generalized waveguide variant; shallow water with a thermocline

## 1. Introduction

Source depth discrimination in shallow water has significant research value, aiming to distinguish surface sources from submerged ones rather than calculating depth. The distinction between these two classes of sources is based on their respective mode spectrum excitation patterns. It exploits the difference in energies of low-order normal modes (also known as trapped modes [1], TMs) and high-order normal modes (non–trapped modes, NTMs), since the surface source cannot excite TMs due to their evanescent mode amplitudes near the surface [2]. In contrast, a submerged source can excite both TMs and NTMs.

Publications have explicitly used the numerical representation of the energy difference for source depth discrimination. A horizontal line array (HLA) at the endfire was utilized to build the mode subspace projections and estimate the energy ratio between these two groups of modes for discrimination [1], requiring the inputs of an approximate sound speed profile, water depth, and bottom type. Mode filtering was also used to build the trapped energy ratio with an HLA close to the endfire [3]. This demands prior knowledge of the mode characteristics, which cannot be precisely obtained, due to the uncertainty in the acoustic model or the environmental mismatch.

The application of the waveguide invariant $\beta$ [4] in depth discrimination implicitly exploits the aforementioned difference, which suggests whether the source is near the surface or submerged, depending on its value. The invariance quantifies the orientation of the intensity striations caused by modal interferences in the frequency–range ($f - r$)

plane. It is found that, for a surface source, the distribution of the $\beta$ peaks are at different values when the receiver is above or below the thermocline [5]. According to the reciprocity principle, the distribution will peak, depending on the depth of the source for a fixed near-surface receiver. $\beta$ can also be extracted by using the warping transform in the frequency domain from the interference pattern without knowing the source range [6]. However, $\beta$ requires relative movement between the source and receiver, since the modal interferences occur in the range domain.

Note that there are modal interferences in the frequency–depth ($f - z$) plane that behave comparably to those in the $f - r$ plane, determined by the dominant modes associated with the source depth. This paper discusses the discrimination between surface and submerged sourcesin shallow water, which are above or below the thermocline, respectively. The group of NTMs, which dominate the surface source, produces an observable striation pattern in the acoustic intensity, while the interferogram of the submerged source is chaotic. The proposed method utilizes a vertical line array (VLA) to "capture" these interference patterns, which are described by a generalized waveguide variant (GWV) distribution. The existence or lack of a distinct peak in the distribution represents the presence or absence of the striation, which is further used to discriminate surface and submerged sources. The method is valid for a higher frequency source, as its NTMs have similar characteristics compared to those from low-frequency sources. The accumulation in the depth domain allows for some robustness against noise. In addition, the value of the GWV is related to the source range; however, the discrimination does not require this value, which means that the source range can be unknown.

The paper is organized as follows. In Section 2, the GWV $\eta$ is derived in the $f - z$ plane with interferometric signal processing. The method based on $\eta$ for source depth discrimination and the requirements of method based on the source frequency are presented. In Section 3, the proposed method is performed on the data from the towing ship in the SWellEx–96 experiment. The striation pattern and $\eta$ of the intensity distribution in the $f - z$ plane generated by the surface source are verified. In Section 4, the numerical modeling results for the same experimental situation are presented. Complementing the simulations of the submerged source, which does not meet the implementation requirement in the experiment, the proposed discrimination method is validated. Furthermore, the performance under noise conditions and for different source ranges is investigated. A summary is presented in Section 5.

## 2. Discrimination Using Intensity Striations in the $f - z$ Plane
### 2.1. Generalized Waveguide Variant Describing the Intensity Striations

For a point source at depth $z_s$, the acoustic intensity at depth $z$ and range $r$ can be expressed as [7]:

$$I(r, z, f, z_s) = \sum_{m=1}^{M} \sum_{n=1}^{M} B_m(r, z_s) \phi_m(z) B_n^*(r, z_s) \phi_n^*(z) e^{i(k_{rm} - k_{rn})r}, \tag{1}$$

where

$$B_m(r, z_s) = \sqrt{2\pi / k_{rm} r} \phi_m(z_s), \tag{2}$$

$M$ is the total number of normal modes excited by the source, and $\phi_m(z)$ and $k_{rm}$ are the depth function and the horizontal wavenumber of the $m$th mode, respectively. $(\cdot)^*$ is the conjugate operator.

The intensity maximum in the frequency–depth ($f - z$) plane is determined by the following condition

$$dI = \frac{\partial I}{\partial z} dz + \frac{\partial I}{\partial f} df = 0. \tag{3}$$

The slope of the striations $\kappa$ is

$$
\begin{aligned}
\kappa = \frac{df}{dz} &= -\frac{\partial I / \partial z}{\partial I / \partial f} \\
&= -\frac{2 \sum\limits_{m=1}^{M} \sum\limits_{n=1}^{M} B_m B_n^* \phi_n^*(z) e^{i(k_{rm}-k_{rn})r} \partial \phi_m / \partial z}{r \sum\limits_{m=1}^{M} \sum\limits_{n=1}^{M} B_m \phi_m B_n^* \phi_n^* e^{i(k_{rm}-k_{rn})r} \left( S_{g,m} - S_{g,n} \right)},
\end{aligned}
\tag{4}
$$

where $S_{g,m}$ is the group slowness of the $m$th mode.

A generalized waveguide variant (GWV) $\eta$ of the $f - z$ plane is defined as

$$
\eta(z, f | z_s) = \kappa * \frac{r}{2} = -\frac{\sum\limits_{m=1}^{M} \sum\limits_{n=1}^{M} B_m B_n^* \phi_n^* e^{i(k_{rm}-k_{rn})r} \partial \phi_m / \partial z}{\sum\limits_{m=1}^{M} \sum\limits_{n=1}^{M} B_m \phi_m B_n^* \phi_n^* e^{i(k_{rm}-k_{rn})r} \left( S_{g,m} - S_{g,n} \right)}.
\tag{5}
$$

Equation (5) can be rewritten as a sum of weighted components $\eta_{mn}$ with coefficients $\alpha_{mn}$ representing their contributions to the variant (similar to the derivation of $\beta$ in [8]), given by

$$
\eta(z, f | z_s) = \sum_{m=1}^{M} \sum_{n=1}^{M} \alpha_{mn} \eta_{mn},
\tag{6}
$$

where

$$
\alpha_{mn} = \frac{B_m \phi_m B_n^* \phi_n^* e^{i(k_{rm}-k_{rn})r} \left( S_{g,m} - S_{g,n} \right)}{\sum\limits_{m=1}^{M} \sum\limits_{n=1}^{M} B_m \phi_m B_n^* \phi_n^* e^{i(k_{rm}-k_{rn})r} \left( S_{g,m} - S_{g,n} \right)},
\tag{7}
$$

$$
\eta_{mn}(f, z) = \frac{-\partial \phi_m / \partial z}{\phi_m \left( S_{g,m} - S_{g,n} \right)}.
\tag{8}
$$

Under the WKB [7] approximation, the mode function $\phi_m(z)$ can be expressed as

$$
\phi_m(f, z) = \sin[k_{zm}(z)z],
\tag{9}
$$

where $k_{zm}(f, z) = \sqrt{[2\pi f / c(z)]^2 - k_{rm}^2}$ is the vertical wavenumber of the $m$th mode. Consider the situation when the VLA is deployed below the thermocline, which means the receivers are below all the turning points of trapped modes, making $c(z)$ and $k_{zm}$ both constants. Therefore,

$$
\eta_{mn}(f, z) = -\frac{k_{zm} \cot(k_{zm}z)}{S_{g,m} - S_{g,n}}.
\tag{10}
$$

Since many $\eta_{mn}$ values, as pairs of modes $(m, n)$, contribute to the GWV, a distribution of $\eta$ denoted by $E_\eta$ better quantifies this complex striation pattern, similar to $E_\beta$ proposed in [9–11].

As mentioned above, the GWV is converted from the slope of the interference striations, which can be calculated by using two-dimensional Fast Fourier Transform [12] (2D–FFT) on the intensity distribution $I(f, z)$. The corresponding algorithm follows [9,10] and will be briefly reviewed below.

The 2D–FFT of $I(f, z)$ with depth aperture $D$ and bandwidth $B$ is defined by

$$
I(x, y) = \left| \int_{f_m - B/2}^{f_m + B/2} \int_{z_m - D/2}^{z_m + D/2} I(f, z) e^{-i2\pi(xz+yf)} dz df \right|,
\tag{11}
$$

where $f_m$ and $z_m$ are the mean values of axis $f$ and $z$, and $x$ (in m$^{-1}$) and y (in s) are the FFT variables conjugate to the depth and frequency, respectively. We replace the slope in Equation (5) with its expression in the Fourier domain and obtain

$$\eta = -\frac{r}{2}\frac{x}{y},\tag{12}$$

and the GWV can be represented in another set of variables related to the polar coordinate system by choosing

$$K = \sqrt{x^2 + y^2}.\tag{13}$$

The GWV distribution $E_\eta$ is given by summing up $K$ in the $(\eta, K)$ plane, which is the result of the polar coordinate transform. The presence of a clear peak indicates the existence of striations and the corresponding slope at $\eta$.

*2.2. Discrimination Based on the GWV*

In shallow water with a thermocline, when the source is located above the thermocline, only the first several NTMs exist, since the TMs are poorly excited, and the higher order modes attenuate rapidly during the propagation. These NTMs can be regarded as a group of modes due to their similar $k_z$s, which will further behave similarly in $\cot(k_z z)$ and $\eta_{mn}$. As long as the sample locations are not at the depths where the $\eta_{mn}$ approaches 0, the enhancement of these $\eta_{mn}$ provides a proper value of $\eta$ (a peak of the GWV distribution) and corresponding striations in the $f - z$ plane.

However, for the submerged source, which excites both TMs and NTMs, the $k_z$s of the modes vary by an order of magnitude, resulting in the change in the function period. Therefore, the sum of $\eta_{mn}$ containing $\cot(k_z z)$ with different periods cannot make $\eta$ a certain value, which means that the interferogram will be chaotic, and no striations will exist.

As a crucial parameter, shown in Equation (10), for the proposed method, $k_{zm}$ determines $\eta_{mn}$ and $\eta$ for the fixed receiver depths. In general, $k_z$ increases with the source frequency, but its rate decreases. For a higher-frequency surface source, the $k_z$s of NTMs vary slowly during the frequency band, resulting in similar periods of $\cot(k_z z)$, which further ensure the enhancement of those $\eta_{mn}$. In the case of discriminating the lower-frequency surface source, the fact that the $k_z$s of NTMs in the processing band vary greatly, and the periods of $\cot(k_z z)$ change rapidly, means the summation of $\eta_{mn}$ is like that of the submerged source, and it fails to distinguish between these two source classes.

As shown in Equations (5) and (12), $\eta$ is related to the source–array range $r$, which implies that the slope of the striation is scaled up/down by the ratio of the range (if the range is estimated before or later, which is outside the scope of this paper). However, the scale change does not affect the striation's existence. It is the fact that the distinct peak of $\eta_{mn}$ exists and not the value itself that is an important clue to whether the target is on the surface or submerged. These above observations are verified using experimental and simulated data in the following sections.

**3. Experimental Data Analysis**

The SWellEx–96 experiment [13] was conducted near San Diego, CA, in May of 1996. The SSP can be approximately regarded as a typical downward refracting profile with a thermocline. The VLA was deployed from a depth of 94.125 m to a depth of 212.25 m and contained 21 elements that were evenly spaced (ignoring the small vertical tilt). The range in the depth was nearly the same as in the situation discussed in Section 2, since it was below the thermocline. The SSP and the array configuration are presented in Figure 1.

One second of data were analyzed, which was 73 min after the start of event S5, where the towing ship (R/V Sproul) was 2.323 km from the VLA. The analyzed data involved one signal with the band $\Delta f = 150$ Hz (600–750 Hz) radiated by the towing ship (at a depth of 2.9 m [14]), which was regarded as a surface source. The reasons for not choosing the two experiment sources were as follows: (i) a broadband source was required, since

it had the premise of an interference structure, while the shallow source transmitted nine frequencies between 109 Hz and 385 Hz; (ii) although the deep source stopped projecting CW tones and started projecting FM chirps (200–400 Hz) at the beginning, midway point, and end of the track, its frequency was not applicable (too low) in this scenario.

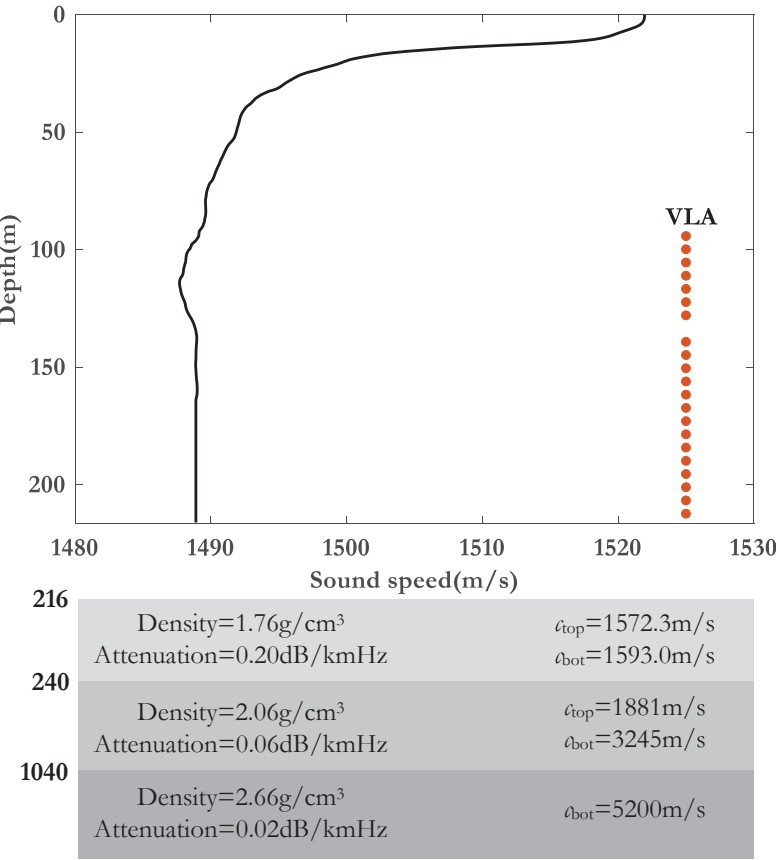

**Figure 1.** The SSP of SWellEx–96 and the arrangement of VLA.

The results of the experimental data processing are shown in Figure 2. The intensity distribution $I(f,z)$ Equation (1), 2D–FFT of $I(f,z)$ Equation (12), 2D–FFT in the polar coordinates, and the GWV distribution $E_\eta$ are shown for the assessing procedure.

One can observe the intensity striations in Figure 2a (to show the striations more clearly, we show the image with a larger bandwidth (550–900 Hz), which is symmetrical at about $f = 750$ Hz, since the sample frequency of the data is 1500 Hz). It is worth mentioning that the striations were not caused by frequency shifting, although there were several tones (such as 605 Hz and 677 Hz) projected by the towing ship, since the speed of the ship was 2.5 m/s (5 knots), and the length of the data was 1s.

Figure 2b shows the result of the 2D–FFT of the region enclosed by the white dashed lines in Figure 2a and exhibits a vertical line resulting from the background noise.

We removed this vertical line and performed the polar coordinate transform (the transformations later were all conducted after vertical line removal) in Figure 2c. Figure 2d shows the GWV distribution $E_\eta$ of the data we analyzed, and the peak $\eta_{ex} = 68.4$.

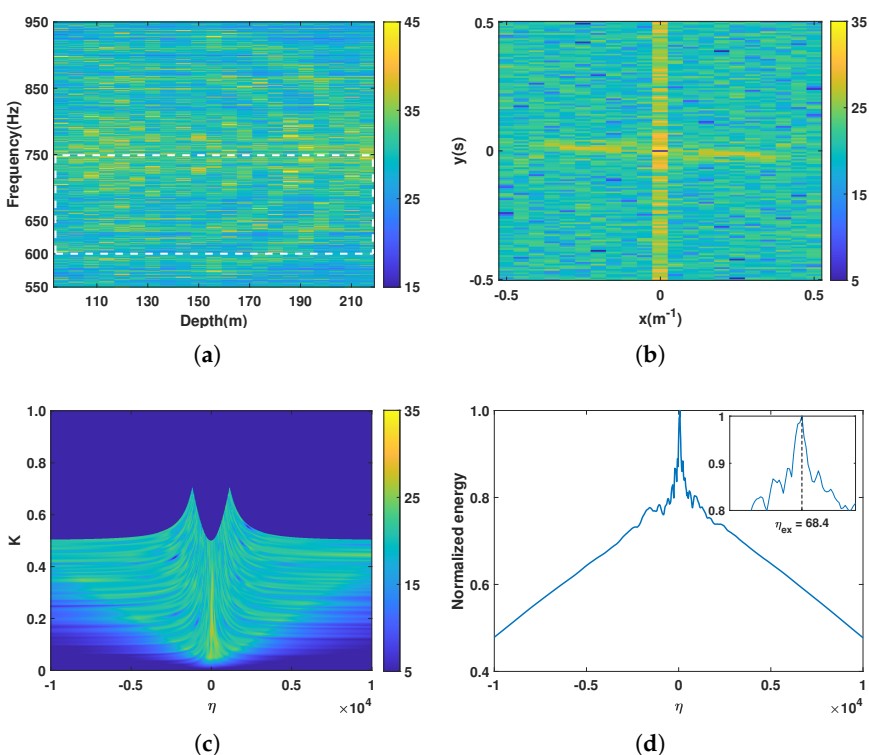

**Figure 2.** The results of the experimental data: (**a**) $I(f,z)$; (**b**) 2D−FFT of $I(f,z)$; (**c**) 2D−FFT in the polar coordinate; (**d**) the GWV distribution $E_\eta$ and zoom of peak.

## 4. Numerical Modeling Results

For numerical simulations, we used the acoustic environment in SWellEx–96, considering the same VLA as that deployed during the experiment. The source frequency band was the same as noted above, $\Delta f = 150$ Hz (600–750 Hz). The horizontal distance between the source and the VLA was 2.323 km. The KRAKEN [15] was used to calculate the pressure field.

### 4.1. The Mode Functions, Normalized Amplitudes of Modes, and $\eta_{mn}$

Figure 3a displays the mode depth functions for the central frequency $f_c = 675$ Hz, with black lines marking the depths of 3 m and 54 m.

Figure 3b,c show the normalized amplitudes of the normal modes excited by a surface source and a submerged source, respectively, with marked TMs and NTMs. As can be seen, the two source classes differ in the dominance of excited modes, providing the basis for depth discrimination.

Figure 3d,e show $\eta_{43,44}$ (typical dominant interference modes of the surface source) as a function of the frequency and the water depth, and $\eta_{43,44}$ versus depth for $f_c = 675$ Hz, with red circles representing the VLA receivers, respectively. One can note that the $\eta_{mn}$ of 600–750 Hz show periodicity and share a similar period, since there is $\cot(k_{zm}z)$ in $\eta_{mn}$, and $k_{zm}$ varies slowly during the processing bandwidth, which shows the potential to make their combination $E_\eta$ have a sharp peak. The VLA receivers are mostly not located near the zero point of $\eta_{43,44}$ versus depth for $f_c = 675$ Hz, letting $\eta$ avoid being 0. The high $\eta_{43,44}$ (the dazzling line in $f = 746$ Hz) in Figure 3d is due to the tiny difference (0.0274 m/s) in group speeds of the 43rd and 44th modes, which happen to be the first two NTMs.

Figure 3f,g show $\eta_{9,10}$ with a larger period, which exhibited an abnormal situation at depths of 115–130m, caused by $c(z)$, leading to unusual $k_{zm}$ and $\cot(k_{zm}z)$, and $\eta_{9,10}$ versus depth for $f_c = 675$ Hz with red circles representing the VLA receivers, respectively. More importantly, for the submerged source that excited both TMs and NTMs, the period of $\cot(k_{z9}z)$ (seen in the Figure 3g) was nearly four times that of $\cot(k_{z43}z)$ or several times

that of the other $\cot(k_{zm}z)$, which means that each type of interference mode contributes its own peak, resulting in multiple sidelobes in the distribution $E_\eta$ .

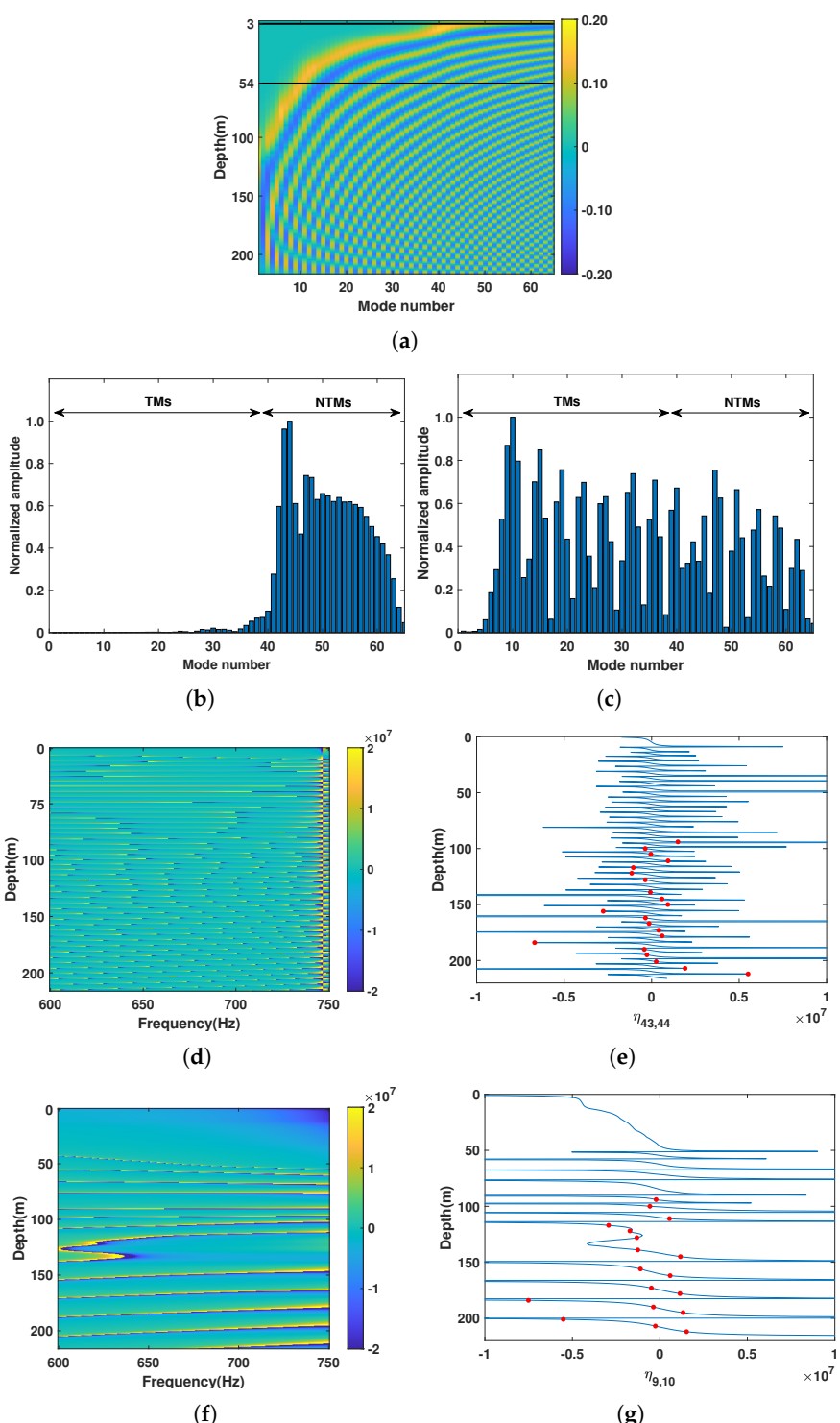

**Figure 3.** Mode depth functions for $f_c$=675Hz (**a**); normalized amplitude of the normal modes excited by the surface/submerged source (**b**,**c**); $\eta_{43,44}$ and $\eta_{9,10}$ as a function of the frequency and the water depth (**d**,**f**); $\eta_{43,44}$ and $\eta_{9,10}$ versus depth for $f_c$=675Hz with red dots representing the values of $\eta_{m,n}$ at the depths of the VLA receivers, respectively (**e**,**g**).

Table 1 presents the total number of modes, the dominant interference modes, and the corresponding period of $\cot(k_{zm}z)$ for different surface source frequencies. In underwater acoustics, in a general sense, a frequency above 500 Hz can be referred to as a high

frequency (mid-high frequency). In the scenario discussed here, it needs to be discussed in combination with the specific SSP, such as shown in Table 1. Under the same bandwidth, the period of the dominant interference at different frequencies changes with a smaller period (600–750 Hz, $5.26/4.42 * 100\% \approx 119\%$) is called higher-frequency, and the opposite is called lower-frequency (150-300 Hz, and 300-450 Hz). There is no absolutely clear boundary between higher and lower frequencies here. For the surface source with lower frequencies (for example, 150–450 Hz), the period of $\cot(k_{zm}z)$ in the dominant interference modes decreases quickly ($15.47/9.52 * 100\% \approx 163\%$, $9.52/6.69 * 100\% \approx 142\%$). This difference between these periods makes the superposition of the NTMs of the surface source behave like those of the TMs and NTMs of the submerged source, and the proposed method fails.

**Table 1.** Total number of modes, the dominant interference modes, and the corresponding period of $\cot(k_{zm}z)$ for different surface source frequencies.

| Surface Source Frequency | 150 Hz | 300 Hz | 450 Hz | 600 Hz | 750 Hz |
|---|---|---|---|---|---|
| Total number of modes | 18 | 32 | 46 | 61 | 73 |
| Dominant interference modes $(m, n)$ | (14,15) | (22,23) | (31,32) | (38,39) | (46,47) |
| Corresponding period of $\cot(k_{zm}z)$ | 15.47 m | 9.52 m | 6.69 m | 5.26 m | 4.42 m |

### 4.2. Performance Study under the SSP from the Experiment

Two cases are considered here: (I) one case of a surface source at a depth of 3 m corresponding to the towing ship; (II) the other of a submerged source at a depth of 54 m.

The interferogram in Figure 4a, 2D–FFT of $I(f, z)$ in Figure 4b, 2D–FFT in the polar coordinates in Figure 4c, and the GWV distribution $E_\eta$ in Figure 4d correspond to case I. Compared to the intensity striations in Figure 2a, those interference structures are more obvious, due to the stationary spectra used in the simulation. Being free from background noise, the vertical line (as in Figure 2b, caused by the noise) disappears. The highest energy of $E_\eta$ denotes the presence of striation, and the number of $\eta_{\text{simu}} = 68.4$ , which is the same as the experimental data result ($\eta_{\text{ex}} = 68.4$). The agreement between the simulation of the surface source and the experimental data analysis proves that there are intensity striations in the $f - z$ plane and verifies the effectiveness of the simulation.

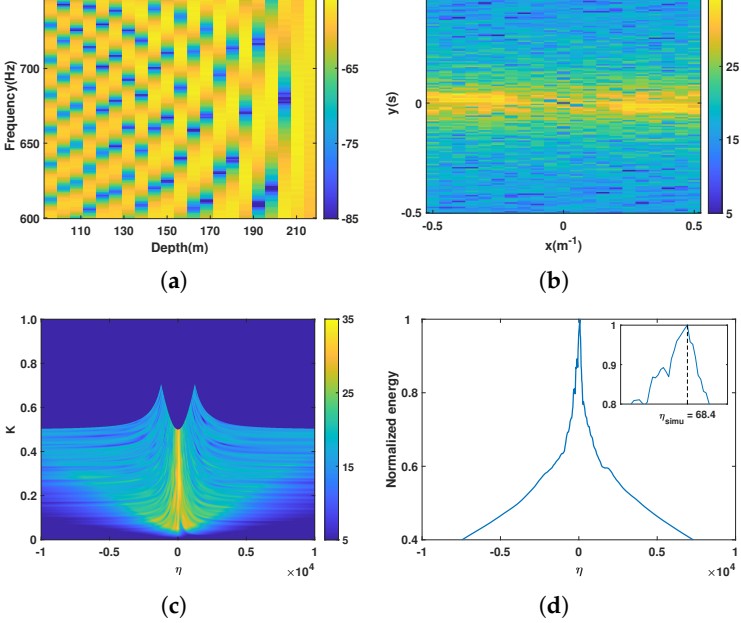

**Figure 4.** The results of the numerical modeling of the surface source (3 m): (**a**) $I(f, z)$ ; (**b**) 2D−FFT of $I(f, z)$; (**c**) 2D−FFT in the polar coordinate; (**d**) the GWV distribution $E_\eta$ and zoom of peak.

The intensity distribution $I(f,z)$, 2D–FFT of it, 2D–FFT in the polar coordinates, and the GWV distribution $E_\eta$ are shown in Figure 5a–d for case II. The interferogram is chaotic. The intensity striation can hardly be found in the picture, let alone its slope. The distribution $E_\eta$ in Figure 5d has many peaks with similar values (differences in the second decimal point), which implies that there are not striations associated with the GWV.

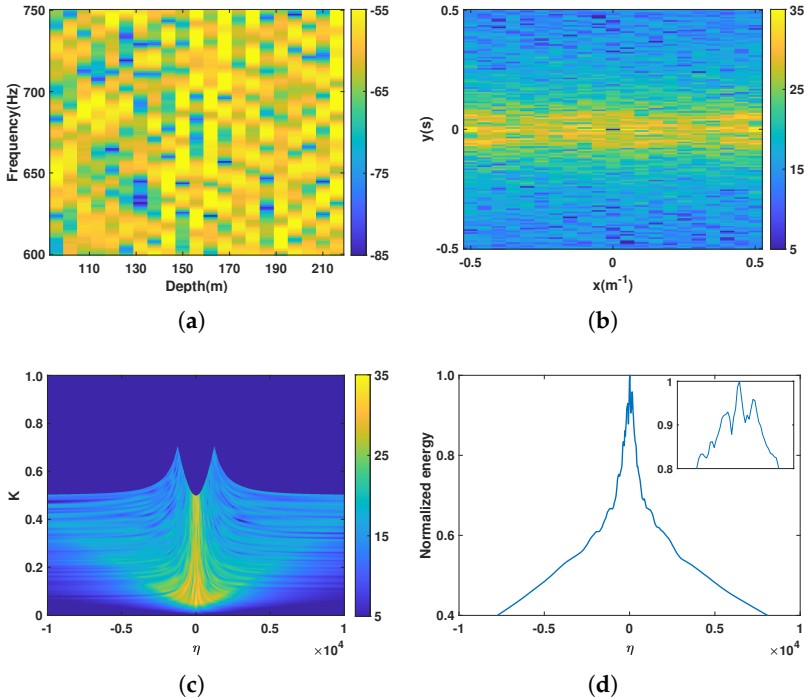

**Figure 5.** The results of the numerical modeling of the submerged source (54 m): (**a**) $I(f,z)$; (**b**) 2D−FFT of $I(f,z)$; (**c**) 2D−FFT in the polar coordinate; (**d**) the GWV distribution $E_\eta$ and zoom of peak.

*4.3. The Effect of the Noise and Source Range on the Performance*

The performance of the proposed method under noise conditions and for different source ranges is described in this section.

We define the signal–noise ratio (SNR) as

$$\text{SNR} = 10\log\left(\frac{\sum_1^L s_l^2}{L}\Big/ \sigma^2\right), \tag{14}$$

where $l$ and $L$ represent the array element index and total number of elements, respectively, $s_l^2$ is the signal power on the $l$th element, and $\sigma^2$ is the noise power.

The results of the processing for two SNRs ($\text{SNR}_1 = -3$ dB and $\text{SNR}_2 = -10$ dB) are shown in Figures 6 and 7, respectively. The results of each polar transform are omitted here.

For the case of $\text{SNR}_1 = -3$dB, there are still observable striations in the intensity distribution of the surface source (Figure 6a), and the peak $\eta_1 = 68.5$ is similar to the $\eta_{\text{ex}} = 68.4$ in the absence of noise. The interferogram of the submerged source (Figure 6d) is still chaotic, and the peak in $E_\eta$ (Figure 6f) cannot be identified.

For the case of $\text{SNR}_2 = -10$ dB, under such noise conditions, the interferograms (Figure 7a,d) are chaotic, and no peak can be identified in either Figure 7c nor Figure 7f. The method works poorly at a low SNR, since its sample aperture is restricted below the thermocline and limited by the water depth, unlike $\beta$ sampling in the $r$ domain for an extendable distance, which enhances the SNR. In addition, it is neither practical nor economical to densely deploy receivers in the $z$ domain.

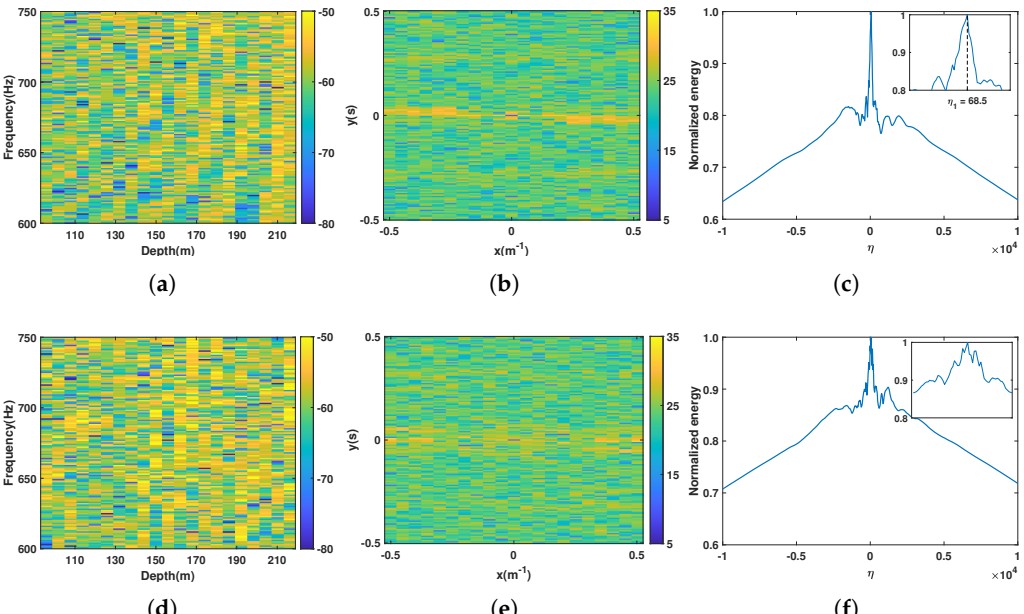

**Figure 6.** The results of the numerical modeling of sources (3 m and 54 m) for $SNR_1 = -3$ dB: $I(f, z)$ (**a**,**d**); 2D−FFT of $I(f, z)$ (**b**,**e**); $E_\eta$ and zoom of peak (**c**,**f**).

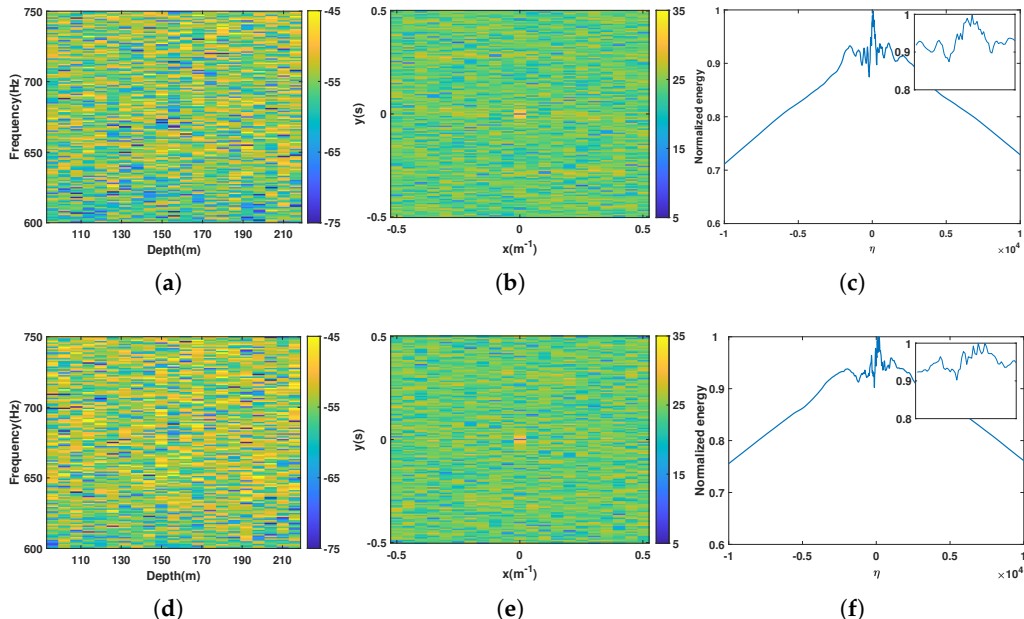

**Figure 7.** The results of the numerical modeling of sources (3 m and 54 m) for $SNR_2 = -10$ dB: $I(f, z)$ (**a**,**d**); 2D−FFT of $I(f, z)$ (**b**,**e**); $E_\eta$ and zoom of peak (**c**,**f**).

The performance for a different source range $r_1 = 3$ km is studied here. Figure 8 shows the results and also omits the polar transform. The interference patterns still exist (Figure 8a), and the peak $\eta_{3km} = 87.9$ is proportional to $\eta_{simu}$, with the ratio between two source ranges ($\eta_{3km}/\eta_{simu} = 87.9/68.4 \approx 1.29 \approx r_{3km}/r = 3/2.323$). The interferogram of the submerged source (Figure 8d) is chaotic, and the peak in $E_\eta$ (Figure 8f) can not be identified.

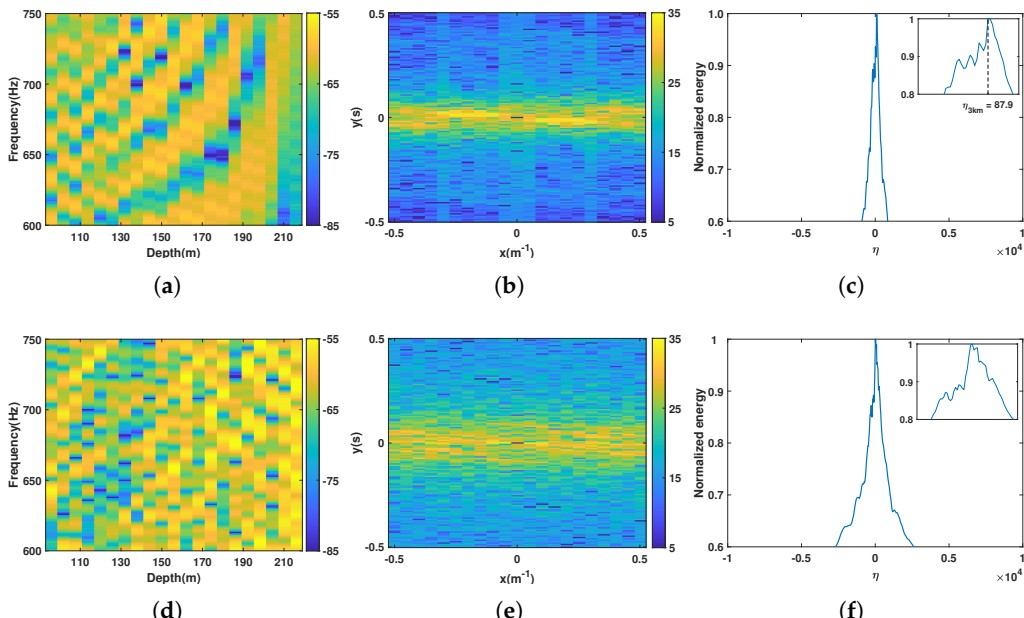

**Figure 8.** The results of the numerical modeling of sources (3 m and 54 m) for $r_{3km}$: $I(f, z)$ (**a,d**); 2D−FFT of $I(f, z)$ (**b,e**); $E_\eta$ and zoom of peak (**c,f**).

## 5. Conclusions

A method for source depth discrimination is presented for VLA based on the presence of intensity striations extracted from the frequency–depth plane. The orientation of this intensity interference pattern is characterized as a generalized waveguide variant called $\eta$, which was derived in this paper, dominated by different types of normal modes excited by a surface/submerged source. Analytical expressions illustrate that for the higher-frequency surface source, the source interferogram shows the intensity striation patterns clearly and the distribution of $\eta$ peaks associated with the source range. However, for the submerged source, the interferogram is chaotic, and the distribution of $\eta$ does not show the peak (there are many high sidelobes).

This method was verified with experimental data and simulated data with reasonable success. For the surface source, there is a good agreement between the experimental intensity striation patterns and those predicted by the theory, as well as the peaks of $\eta$ in each situation. The successful discrimination with a low noise background and different source ranges further indicates the potential of the method on real data. It should be pointed out that, although $\eta$ is related to the source–array range $r$, it is the presence of the striations, not the value of its slope, that we use to determine the depth class of the source.

**Author Contributions:** Conceptualization, X.L.; methodology, X.L. and C.S.; validation, X.L.; formal analysis, X.L.; investigation, X.L.; resources, C.S.; data curation, X.L.; writing—original draft preparation, X.L.; writing—review and editing, C.S.; supervision, C.S. All authors have read and agreed to the published version of the manuscript.

**Funding:** This research received no external funding.

**Data Availability Statement:** The data presented in this study are available on request from the corresponding author.

**Acknowledgments:** The authors would like to thank the SWellEx-96 team for making the experiment data publicly available.

**Conflicts of Interest:** The authors declare no conflicts of interest.

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
