# Peer review of "Source Depth Discrimination Using Intensity Striations in the Frequency–Depth Plane in Shallow Water with a Thermocline"

_remotesensing, doi:10.3390/rs16040639_

Round 1

Reviewer 1 Report

Comments and Suggestions for Authors

This paper is considered a good paper as a technical note paper, with some notes:

1/ There is a selective consideration of the processed data.

2/ It is a distinct subject more closely related to the specialization of agnostic and signal processing.

3/ I put some comments enclosed in the yellow box above the text.

Comments on the Quality of English Language

Reviewer 2 Report

Comments and Suggestions for Authors

The only issue is how situation in clutter noisy environments? Because noisy measurements can be solved in this paper, but, if the input also contains biases and clutter in experiments, what is the robustness of the proposed paper? How strong could be supported in true measurements, regards the rate in all measurements. 

Comments on the Quality of English Language

Nice paper.  

Reviewer 3 Report

Comments and Suggestions for Authors

The manuscript proposes a method for source depth discrimination, based on intensity striations in the frequency-depth plane with a vertical linear array in a shallow water environment; the method is further discussed considering experimental data from the SWellEx–96 experiment, which was collected on a Vertical Line Array; the waveguide was characterized by a thermocline extremely close to the surface. The method is based on the generalized waveguide variant (GWV), which is expected to exhibit clear striations for a surface source, while for a submerged source the pattern of striations becomes diffuse and chaotic. The reviewer believes that the manuscript has a merit, but can not recommend it for publication because the discussion seems to be incomplete. To be more specific the following issues need to be addressed:
- Mode excitation, as described in the manuscript, is specific to environments with a thermocline; thus it is unclear how the method will perform in environments with "smooth" sound speed profiles.
- Line 100: "x (in m-1 ) and y (in s) are FFT variables conjugate to depth and frequency, respectively." Then, what's the meaning of y' in Eq(12)?
- Line 106: "The highest energy denotes the presence of striation slope at eta." Energy? Perhaps is not energy but amplitude? The statement is also ambiguous, perhaps what the authors mean is that the presence of a clear peak indicates the existence of striations?
Line 196: "...source with lower frequencies (for example, 150–450Hz), the period of cot(k zm z) of the dominant interference modes decreases quickly (15.47/9.52 ∗ 100% = 163%, 9.52/6.69 ∗ 100% = 142%)." Values above 100% seem meaningless, choose another reference.
- It is rather open to discussion the statement that the interferogram in Figure 5(a) is chaotic, when compared with the interferogram in Figure 4(a). It might appear visually chaotic to a researcher, but it is unclear how to measure the chaos with an algorithm. Besides, as presented, Figures 4(d) and 5(d) show bery little difference, and it is unclear if eta is not peaked again at 68.4. A similar problem happens when comparing Figures 6(c) and 6(f).

Recommendations:
- please display the figure items ((a),(b),(c),...) over the figures, not below them.
- Use the same sign of Transmission Loss for Figures 2(a) and 4(a).

Comments on the Quality of English Language

The writing style requires minor improvements.

Reviewer 4 Report

Comments and Suggestions for Authors

The comments are in attachment.

Round 2

Reviewer 4 Report

Comments and Suggestions for Authors

In attachment
